**DOI: 10.1038/ncomms13131**　　**OPEN**

# Molecular analysis of aggressive renal cell carcinoma with unclassified histology reveals distinct subsets

Ying-Bei Chen[1], Jianing Xu[2], Anders Jacobsen Skanderup[3,†], Yiyu Dong[2], A. Rose Brannon[1], Lu Wang[1], Helen H. Won[1], Patricia I. Wang[2], Gouri J. Nanjangud[4], Achim A. Jungbluth[1], Wei Li[5], Virginia Ojeda[5], A. Ari Hakimi[6], Martin H. Voss[7], Nikolaus Schultz[3], Robert J. Motzer[7], Paul Russo[6], Emily H. Cheng[1,2], Filippo G. Giancotti[5,†], William Lee[3,8], Michael F. Berger[1,2], Satish K. Tickoo[1], Victor E. Reuter[1] & James J. Hsieh[2,7,9]

Renal cell carcinomas with unclassified histology (uRCC) constitute a significant portion of aggressive non-clear cell renal cell carcinomas that have no standard therapy. The oncogenic drivers in these tumours are unknown. Here we perform a molecular analysis of 62 high-grade primary uRCC, incorporating targeted cancer gene sequencing, RNA sequencing, single-nucleotide polymorphism array, fluorescence *in situ* hybridization, immuno-histochemistry and cell-based assays. We identify recurrent somatic mutations in 29 genes, including *NF2* (18%), *SETD2* (18%), *BAP1* (13%), *KMT2C* (10%) and *MTOR* (8%). Integrated analysis reveals a subset of 26% uRCC characterized by NF2 loss, dysregulated Hippo–YAP pathway and worse survival, whereas 21% uRCC with mutations of *MTOR*, *TSC1*, *TSC2* or *PTEN* and hyperactive mTORC1 signalling are associated with better clinical outcome. FH deficiency (6%), chromatin/DNA damage regulator mutations (21%) and ALK translocation (2%) distinguish additional cases. Altogether, this study reveals distinct molecular subsets for 76% of our uRCC cohort, which could have diagnostic and therapeutic implications.

[1] Department of Pathology, Memorial Sloan Kettering Cancer Center, New York, New York 10065, USA. [2] Human Oncology and Pathogenesis Program, Memorial Sloan Kettering Cancer Center, New York, New York 10065, USA. [3] Computational Biology Program, Sloan Kettering Institute for Cancer Research, Memorial Sloan Kettering Cancer Center, New York, NY 10065, USA. [4] Molecular Cytogenetics Laboratory, Memorial Sloan Kettering Cancer Center, New York, New York 10065, USA. [5] Cell Biology Program, Sloan Kettering Institute for Cancer Research, Memorial Sloan Kettering Cancer Center, New York, New York 10065, USA. [6] Department of Surgery, Urology Service, Memorial Sloan Kettering Cancer Center, New York, New York 10065, USA. [7] Department of Medicine, Genitourinary Oncology Service, Memorial Sloan Kettering Cancer Center, New York, New York 10065, USA. [8] Department of Radiation Oncology, Memorial Sloan Kettering Cancer Center, New York, New York 10065, USA. [9] Department of Medicine, Weill Cornell Medical College, 1300 York Ave, New York, New York 10065, USA. † Present addresses: Computational and Systems Biology, Genome Institute of Singapore, 60 Biopolis St, Singapore 138672, Singapore (A.J.S.); Department of Cancer Biology and David H. Koch Center for Applied Research of Genitourinary Cancers, The University of Texas MD Anderson Cancer Center, 1515 Holcombe Boulevard, Houston, Texas 77030, USA (F.G.G.). Correspondence and requests for materials should be addressed to Y.-B.C. (email: cheny@mskcc.org) or to J.J.H. (email: hsiehj@mskcc.org).

Renal cell carcinoma (RCC) encompasses a heterogeneous group of tumours and is mainly categorized based on unique histopathological features. Major subtypes are clear cell RCC (ccRCC, ~75%), papillary RCC (pRCC, ~15%) and chromophobe RCC (chRCC, ~5%)[1–3]. uRCC accounts for 4–5% of RCC that is not classifiable as one of the major (>5%) or the rare (<1%) subtypes such as medullary, collecting duct, mucinous tubular and spindle cell carcinoma, and MiTF family translocation RCC[2,3]. uRCC represents a large proportion of metastatic RCC that exhibits non-clear cell histology (nccRCC), has no standard therapy[4–6], and presents formidable diagnostic and management challenges[7–9]. Large collaborative genomic efforts, including The Cancer Genome Atlas projects, have greatly extended our molecular understanding of common RCC subtypes, including ccRCC[10–13], chRCC[14,15] and pRCC[15–17]. However, as a rare and heterogenous group of tumours, uRCC currently remains as the largest molecularly uncharacterized RCC category with unknown oncogenic pathways.

To gain knowledge towards this unmet need in the diagnosis and management of aggressive nccRCC, we conducted the first in-depth molecular characterization of uRCC in a cohort of 62 primary tumours with high-grade histologic features, all of which were re-reviewed by experienced genitourinary pathologists to ensure their proper classification based on the current World Health Organization and International Society of Urologic Pathology consensus diagnostic criteria[2,3]. To study the spectrum of this heterogeneous group of tumours and not to exclude cases with only formalin-fixed, paraffin-embedded (FFPE) archival tissue, we employ an integrated and step-wise approach, combining targeted cancer gene sequencing, RNA sequencing (RNA-seq), single-nucleotide polymorphism (SNP) array, fluorescence in situ hybridization (FISH), immunohistochemistry and cell-based assays to focus on identifying molecular alterations and pathways that are potentially clinically informative. We find recurrent somatic mutations in 29 genes, and identify distinct molecular subsets that are characterized by NF2 loss, hyperactive mTORC1 signalling, FH deficiency, chromatin/DNA damage regulator mutations or ALK translocation and associated with varying clinical outcomes.

## Results

**Mutation landscape of uRCC by targeted gene sequencing.** The clinicopathologic features and outcomes of this 62-patient uRCC cohort are summarized in Supplementary Table 1. At the time of nephrectomy, 58% of cases were locally advanced (pT3 and above), with 32% showing regional lymph node involvement. Overall, 42% ($n = 26$) of patients developed metastatic disease and 35% ($n = 22$) died of RCC, underscoring the aggressive clinical behaviour and poor response to systemic therapies observed in this uRCC cohort.

To investigate the molecular aberrations in uRCC, we first employed the Integrated Mutation Profiling of Actionable Cancer Targets (IMPACT) assay, a customized ultra-deep targeted next-generation sequencing platform designed to capture all exons and selected introns of 230 oncogenes, tumour suppressor genes, and members of pathways deemed actionable by targeted therapies (Supplementary Data 1)[18,19]. We identified 29 recurrently mutated genes with an average of 2.6 (0–8) protein-coding somatic mutations per patient tumour (Fig. 1; Supplementary Data 2). NF2 (18%), SETD2 (18%) and BAP1 (13%) were the three most frequently mutated genes. The incidence of NF2 mutations in our cohort is markedly higher than what is reported in ccRCC (0–1%)[10,11,20], pRCC (0–6%)[15–17] and chRCC (0%)[14,15]. In ccRCC, VHL mutations occur at ~75%, and SETD2 and BAP1 at 10–20% frequencies[21], whereas in our uRCC

cohort, only a single VHL mutation was detected in one case (T08). There were 13 genes mutated at 5–10%, among which 5 are epigenetic regulators: KMT2C (10%), KMT2D (5%), ATRX (7%), DNMT3A (5%) and SMARCB1 (5%); 4 are mTORC1 pathway regulators: mTOR (8%), TSC1 (7%), TSC2 (5%) and PTEN (7%); and 3 are transcription factors: KLF6 (5%), NOTCH2 (5%) and TP53 (5%). Four cases only harboured mutations in non-recurrently mutated genes, whereas no mutations were detected in nine cases (15%; Supplementary Data 2).

**uRCC with NF2 loss and dysregulated Hippo–YAP signalling.** The enrichment of cases with NF2 mutations (11 of 62) discovered in our uRCC cohort suggests that NF2 loss could potentially define a molecular subset of uRCC. To assess the NF2 status in uRCC beyond mutations, we next assessed the status of chromosome 22q12 where NF2 resides. Based on copy-number plots generated by the IMPACT sequencing, 22q12 loss was evident in 14 cases (23%), among which 9 also carried NF2 mutations. High-resolution, genome-wide SNP array analysis was performed for 15 of the 16 uRCC cases carrying NF2 mutations and/or exhibiting 22q12 copy-number loss (referred to as the 'NF2 loss' subset from here onwards; Fig. 2a,b; Supplementary Fig. 1). Thirteen cases were confirmed to exhibit hemizygous loss of 22q and the remaining two tumours (T22 and T64), known to carry NF2 somatic mutations, showed copy-neutral loss of heterozygosity (LOH) of 22q (Supplementary Fig. 1). Of note, frequent, concurrent NF2 somatic mutation and chromosome 22q loss has not been reported in RCC. Furthermore, a three-probe FISH assay was performed, which validated the 22q hemizygous loss cases ($n = 14$) within the NF2 loss subset of uRCC (Fig. 2c). Consistent with genomic analyses, the NF2 protein level assessed by immunohistochemistry was significantly lower in the NF2 loss subset than in the remaining uRCC (Fig. 2d).

Germline mutation of NF2 is the principal genetic event underlying the human neurofibromatosis type 2 cancer predisposition syndrome[22,23]. The role of NF2 as a tumour suppressor gene is further demonstrated by mouse models in which genetic loss of Nf2 results in various cancers[24–26]. NF2, a pleotropic factor, plays key roles in cell–cell contact inhibition, growth factor signalling, stem cell and Hippo developmental pathways[22]. We first focused on the NF2–Hippo tumour suppressor network, which was highlighted by a series of reports showing that NF2 enforces the Hippo tumour suppression signalling pathway by phosphorylating, sequestering, degrading and suppressing YAP/TAZ nuclear translocation, thereby disrupting oncogenic transcription[27–32]. To evaluate whether YAP/TAZ is active in subsets of our uRCC cohort, we determined YAP/TAZ protein expression, phosphorylation and intracellular localization by immunohistochemistry. When comparing uRCC with NF2 loss ($n = 16$) to those without ($n = 43$), there was a statistically significant stronger nuclear YAP/TAZ signal, correlating with negative to very low phospho-YAP signal, in NF2 loss tumours (Fig. 2e). To confirm that nucleus-accumulated YAP/TAZ denotes an aberrant YAP/TAZ transcription program, we performed RNA-seq on seven uRCC (four with NF2 loss and three without). Gene Set Enrichment Analysis (GSEA) demonstrated a significant enrichment of an established YAP/TAZ transcription signature[33] in the NF2 loss uRCC (Fig. 2f; Supplementary Data 3). Taken together, these data suggest a novel subset of 16 (26%) uRCC cases with NF2 loss that demonstrates dysregulated Hippo signalling and YAP activation. The importance of YAP/TAZ signalling in NF2 loss kidney cancer was further assessed using ACHN and LB996-RCC cells, two NF2 loss, nccRCC cell lines. Knockdown of YAP in ACHN or

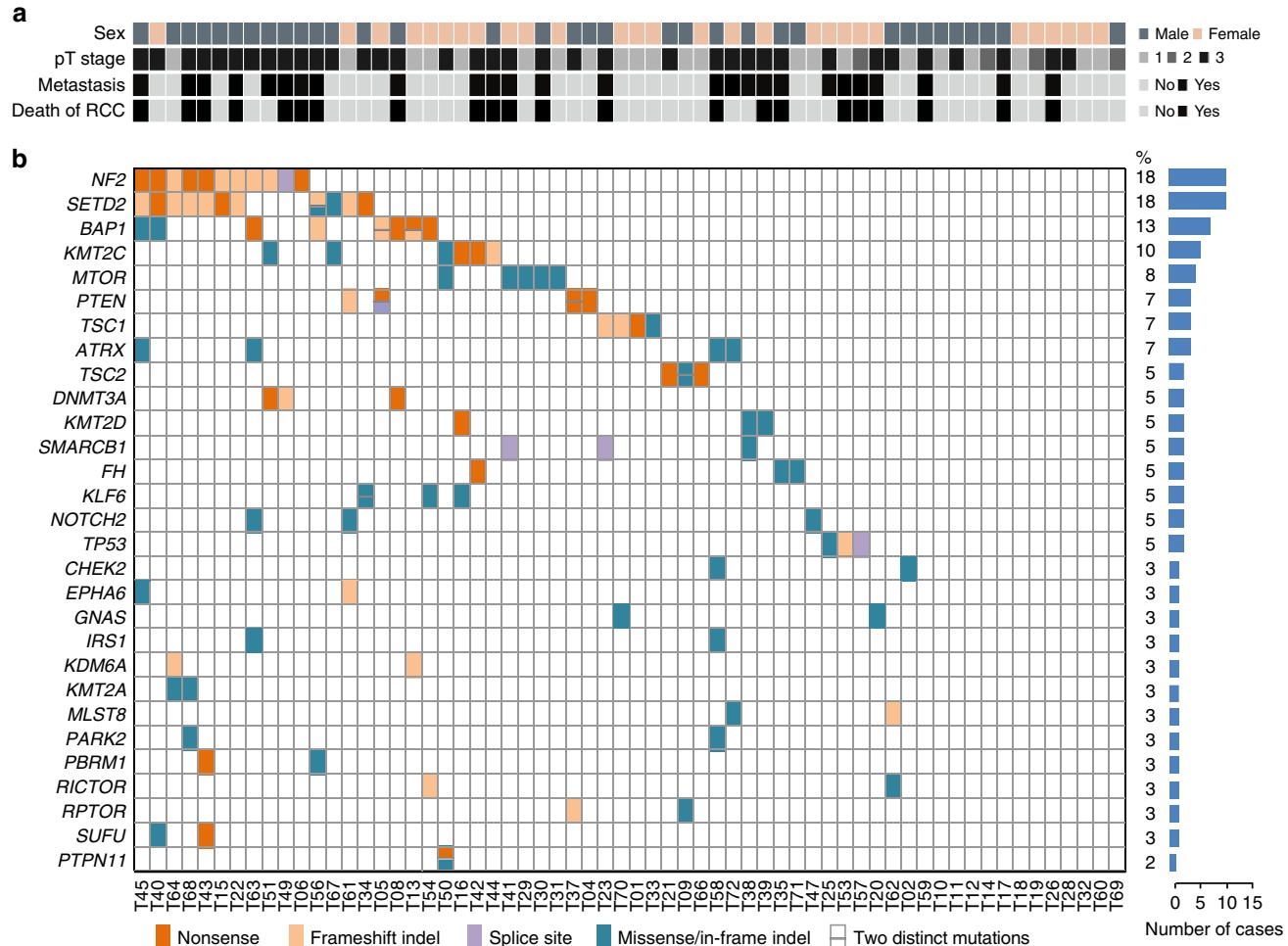

**Figure 1 | Recurrent somatic mutations identified in high-grade uRCC.** (**a**) Key clinicopathological characteristics of our 62-patient uRCC cohort. Pathological (pT) stage was determined according to the 7th edition of the American Joint Committee staging system for renal cancer. The status of metastasis for individual patients was determined at their last follow-up visits or death. (**b**) Mutational landscape of recurrent somatic mutations based on IMPACT assays. Mutated genes are listed on the left, and denoted by individual rows. Sixty-two individual patient tumours are presented as columns and labelled at the bottom (T#). Mutation frequency (%) and absolute number of cases with non-silent mutations detected on individual genes are listed on the right. Mutation frequency was calculated as the percentage of individual tumours with mutation(s) in the indicated genes.

LB996-RCC cells resulted in a decrease of proliferating cells (S and G2/M phases; Fig. 2g; Supplementary Fig. 2), as well as a reduced colony formation in soft agar (Fig. 2h).

Among this NF2 loss subset of uRCC, chromosome 1p and/or 3p losses were also detected in >50% of cases (Fig. 2a,b). Interestingly, while concurrent 3p loss and *VHL* inactivation were reported in ~90% ccRCC, our uRCC with 3p loss did not carry *VHL* mutation or display histologic features of ccRCC. Furthermore, the occurrence of *SETD2* (3p21) mutation was significantly higher in the NF2 loss than in the remaining uRCC tumours (44% versus 9%, Fisher's exact test, $P = 0.004$). *SETD2* encodes a histone H3 lysine 36 (H3K36) methyl transferase. A complete functional loss of SETD2 determined by the respective loss of histone H3K36me3 mark was detected in all seven NF2 loss, *SETD2*-mutated cases (Supplementary Fig. 3). In contrast, 54 of the remaining 55 uRCC tumours retained the H3K36me3 mark. Recurrent mutations of the other chromatin modulating genes including *BAP1* did not show significant enrichment in the NF2 loss subset.

The NF2 loss uRCC exhibited a range of architectural patterns with multinodular or infiltrative growth (Supplementary Fig. 4). The morphologic spectrum of our NF2 loss uRCC did not fulfill diagnostic criteria of type 2 pRCC or collecting duct RCC[2,3]. Nevertheless, as small number of RCC with *NF2* mutations have been recently reported in pRCC[15,16] and collecting duct RCC[34], it remains to be determined whether these tumours were distinct from or overlapped with our NF2 loss uRCC.

**uRCC with hyperactive mTORC1 signalling**. Somatic mutation analysis of our uRCC cohort demonstrated that potentially mTORC1 pathway activating mutations comprising *MTOR* (5), *TSC1* (4), *TSC2* (3) and *PTEN* (4) occurred mutually exclusively in 16 (26%) cases, which might indicate another distinct subset (Fig. 3a). Mutations of these genes have been described in ccRCC (12%)[10], pRCC (8%)[16] and chRCC (9%)[14]. Of the *MTOR* mutations seen in this cohort (Fig. 3b), I1973F has been described and shown to be hyperactive in cell-based assays[35,36], whereas L2427R (recurred three times in our uRCC cohort) and V2475M mutations have not yet been reported. To interrogate the functional impact of individual *MTOR* mutations, we generated MTOR L2427R and V2475M mutants, and assessed the mTORC1 activity by phosphorylation of S6K and 4EBP1, two key mTORC1 downstream substrates[37]. When the MTOR

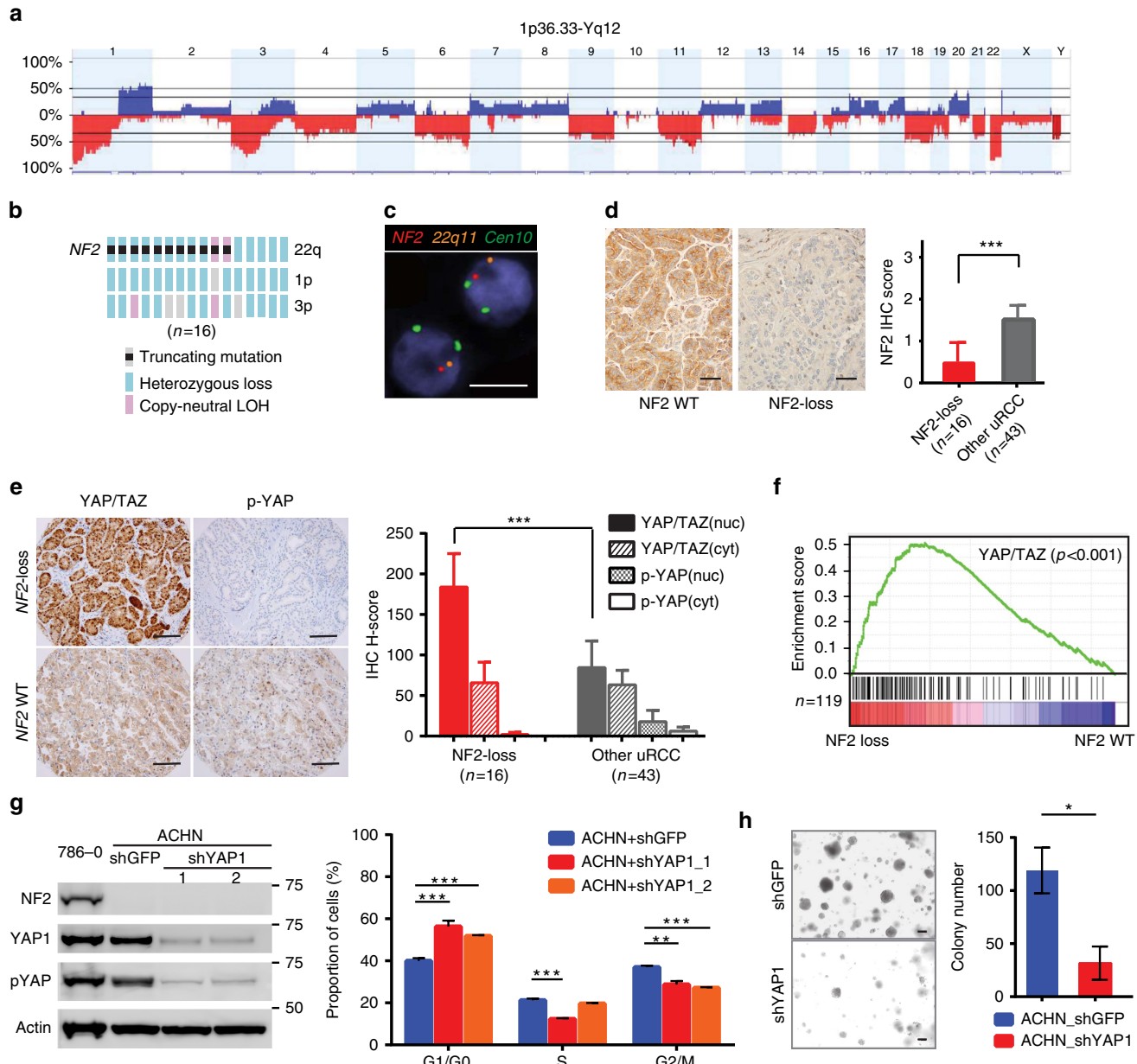

**Figure 2 | Molecular characterizations of the NF2 loss uRCC subset.** (**a**) Genome-wide frequency plot of DNA copy-number gains (blue) and losses (red) across all chromosomes was determined by OncoScan SNP assay in 15 of the 16 uRCC tumours carrying *NF2* mutations and/or 22q loss. The *y* axis denotes frequencies of alteration in individual chromosomal regions. Copy-neutral loss of heterozygosity (CN-LOH) is shown in Supplementary Fig. 1. (**b**) Summary of *NF2* mutations and frequent (>50%) arm-level copy-number alterations (22q, 1p and 3p) detected by sequencing and SNP array analyses of the NF2 loss subset (*n* = 16). Truncating mutations include nonsense mutations, insertions or deletions that alter the reading frame and splice-site mutations. (**c**) Representative hemizygous losses of chromosome 22q and the *NF2* locus were demonstrated by a custom three-probe FISH assay (red, *NF2*; orange, 22q11; green, chromosome 10 centromere). Scale bar, 10 µm. (**d**) Representative immunohistochemical stains of NF2 on NF2 wild-type (WT) and NF2 loss tumours are shown. Scale bars, 50 µm. Semiquantitative IHC scores (0—negative; 1—focal/weak staining; 2—moderate staining; 3—strong and diffuse staining) comparing the NF2 loss subset and the other uRCC tumours are presented as a bar graph. Bars, mean values; error bars, 95% CI. (**e**) Representative images of NF2 WT and NF2 loss uRCC tumours stained by YAP/TAZ and p-YAP antibodies (left panel) are shown. Scale bars, 100 µm. Immunostaining scores (H-scores) for YAP/TAZ and p-YAP nuclear and cytoplasmic staining were determined and presented as a bar graph on NF2 loss (*n* = 16) or other uRCC (*n* = 43) tumours. H-Scores (H = intensity (0–3) × percentage of positive cells (1–100)). Bars, mean values; error bars, 95% CI. (**f**) GSEA plot of the ranked list of differentially expression genes in uRCC with *NF2* loss and those with WT *NF2* generated using a previously established YAP/TAZ-regulated gene set. (**g**) Immunoblots with the indicated antibodies (left) and a bar graph of cell cycle analysis in ACHN cells with YAP1 or control (GFP) knockdown are shown. Bars, mean values; error bars, s.e.m.; replicates *n* = 3. (**h**) Representative images (left) and a bar graph of colony formed in the soft agar after plating $10^5$ YAP1 or control (GFP) knockdown cells are shown. Scale bars, 200 µm. Bars, mean values; error bars, s.e.m.; replicates *n* = 3. Statistical significance was determined by Mann–Whitney *U*-test in **d** and **e**, and by Student's *t*-test in **g** and **h**. Statistical significance is indicated as \***P** < 0.001; \*\***P** < 0.01; \***P** < 0.05.

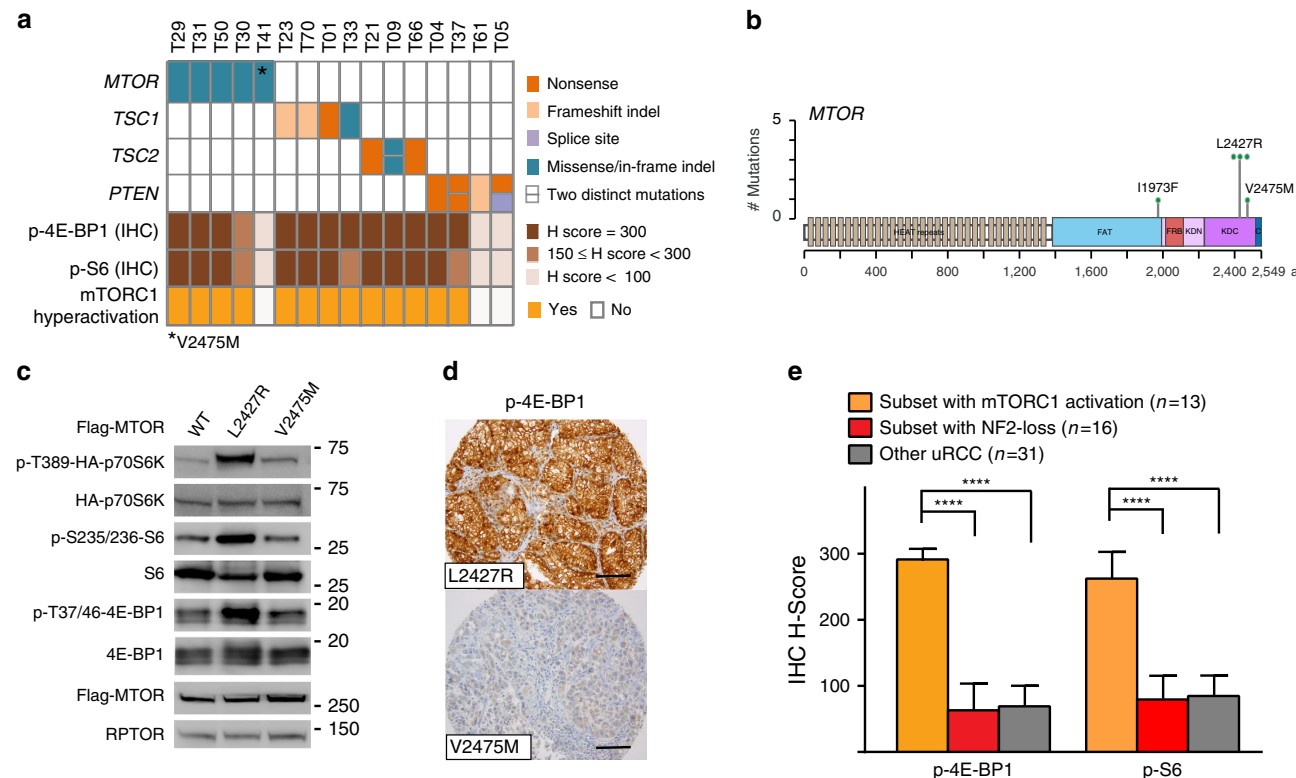

**Figure 3 | uRCC subset with hyperactive mTORC1 signalling. (a)** Schematic overview of indicated mutational and immunohistochemical analyses with annotation of mTORC1 hyperactivation on a subset of uRCC ($n = 16$). **(b)** Depiction of *MTOR* missense mutations identified in 5 (8%) uRCC tumours. MTOR L2427R mutation recurred in three individual tumours. **(c)** Functional analyses of MTOR L2427R and V2475M mutants. 293T cells were transfected with the indicated Flag-tagged MTOR expression constructs in conjunction with HA-S6K. Cellular extracts were collected 48 h later and probed with the indicated antibodies. **(d)** Representative images of p-4EBP1 immunostaining in L2427R and V2475M MTOR mutant tumours. Scale bars, 100 μm. **(e)** Immunostaining scores (H-scores) of p-4EBP1 and p-S6 were determined and presented as a bar graph on mTORC1 hypearctivation ($n = 13$), NF2 loss ($n = 16$) or the other uRCC ($n = 31$) tumours. H-Scores (H = intensity (0–3) × percentage of positive cells (1–100)). Bars, mean values; error bars, 95% CI. Statistical significance was determined by Mann–Whitney U-test. Statistical significance is indicated as ****$P < 0.0001$.

mutant was co-expressed with HA-S6K in 293T human embryonic kidney cells, L2427R exhibited higher activity, whereas V2475M showed baseline mTORC1 kinase activity comparable to the wild-type MTOR (Fig. 3c). Consistent with cell-based assays, immunohistochemistry of the uRCC with L2427R mutation displayed strong p-4EBP1 and p-S6 staining, whereas that of V2475M did not (Fig. 3a,d). These findings suggest that the recurrent I1973F and L2427R MTOR mutations are likely pathogenic, whereas V2475M could be a passenger mutation. Notably, all seven tumours with *TSC1* or *TSC2* mutations had high level of p-4EBP1 (H score = 300), whereas only two of four tumours with *PTEN* mutations exhibited such staining (Fig. 3a). Altogether, our integrated analysis demonstrated that 13 of the 16 uRCC tumours with *MTOR*, *TSC1*, *TSC2* or *PTEN* mutations exhibited hyperactive mTORC1 signals (Fig. 3a,e).

While mTORC1 has been shown to be hyperactive in *NF2*-deficient mesothelioma and meningioma cell lines[38,39], we did not observe hyperactive mTORC1 signalling in NF2 loss uRCC (Fig. 3e). Within our uRCC cohort, the identified NF2 loss (26%) and mTORC1 hyperactive (21%) subsets were mutually exclusive (Fig. 4a).

**Additional molecular subsets detected in uRCC.** As germline and somatic mutations of *FH* have been described in hereditary leiomyomatosis RCC (HLRCC) and a small number of sporadic type II pRCC[6,16,40], we performed 2SC (2-succino-cystein) and

FH immunohistochemistry to investigate the recurrent *FH* somatic mutations observed in three of our uRCC cases (Fig. 1a). FH protein loss and 2SC (detects aberrant protein succination) are highly specific markers for FH-deficient RCC[41–43]. FH and 2SC assays were inversely correlated in our uRCC cohort, and they identified four tumours that were positive for 2SC and negative for FH staining (Supplementary Fig. 5). Genetic testing revealed *FH* germline mutations in three of these four patients, confirming that they indeed represent HLRCC cases. The remaining FH-negative/2SC-positive tumour (T41) harboured somatic homozygous deletion of the *FH* gene, revealing a somatic mechanism that can lead to *FH* functional loss (Supplementary Fig. 5). On the other hand, one tumour (T71) with *FH* G401V somatic mutation was found to be FH positive/2SC negative, and lacked histologic features of HLRCC or FH-deficient RCC[40,42], suggesting that *FH* G401V might be better categorized as a passenger mutation (Fig. 4a; Supplementary Fig. 5).

We also discovered by IMPACT that one uRCC (T12) carried a *TPM3–ALK* fusion, which was further confirmed by FISH analysis (Supplementary Fig. 6). The *TPM3–ALK* fusion has been reported in human cancers[44], including in one uRCC case[45]. Reported rarely in children and adults, ALK translocation-associated RCC is currently considered as an emerging entity, awaiting further characterization[3].

Collectively, these four distinct molecular subsets (NF2 loss, mTORC1 hyperactive, FH-deficient and ALK translocation)

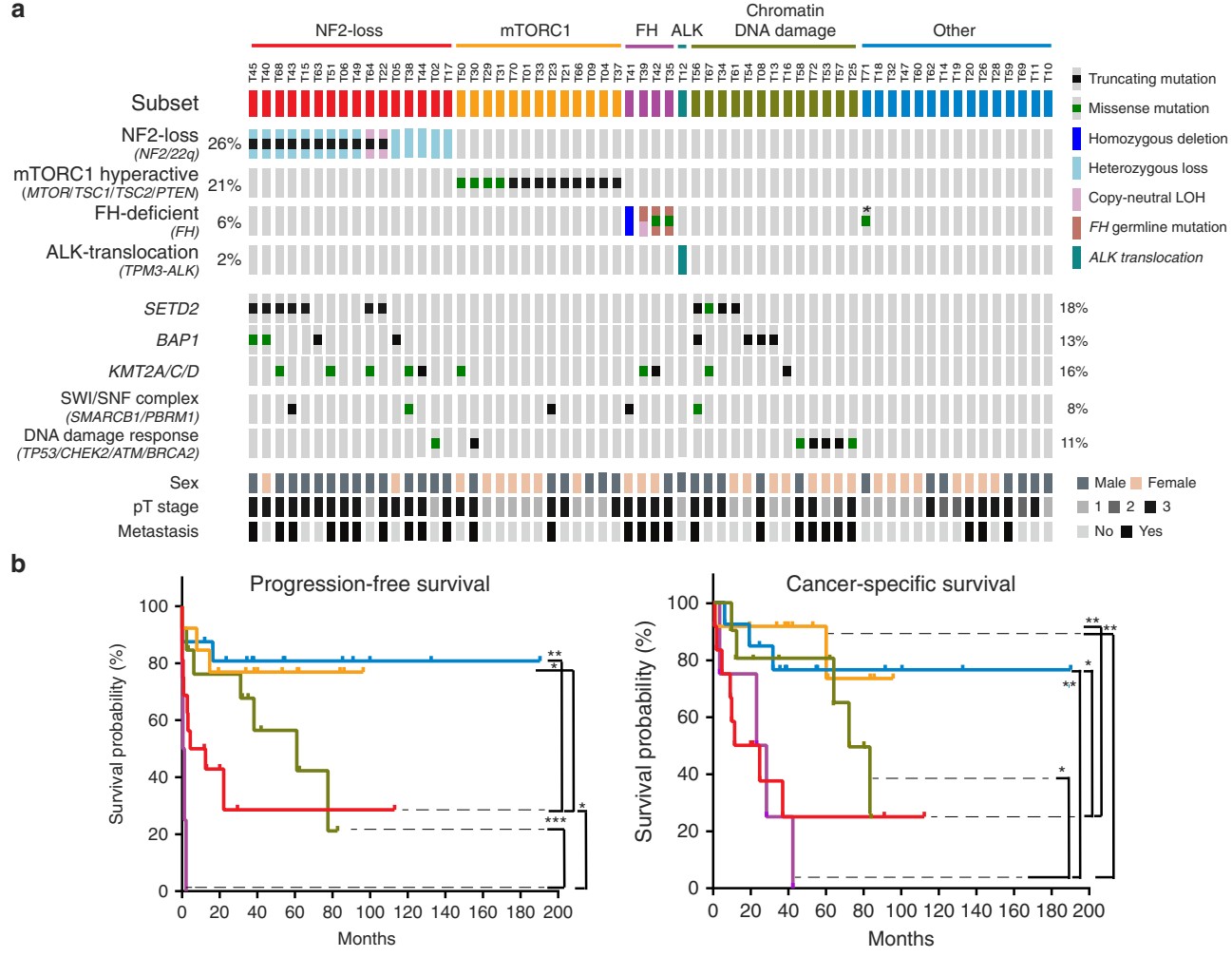

**Figure 4 | Clinical outcomes associated with molecular subsets of uRCC. (a)** Overview of molecular features and clinicopathological characteristics of uRCC subsets identified in our cohort. NF2 loss (NF2 loss, *n* = 16), mTORC1 (mTORC1 hyperactive, *n* = 13), FH (FH deficient, *n* = 4), ALK (*ALK* translocation, *n* = 1), chromatin DNA damage (mutations in chromatin modulation or DNA damage response genes, *n* = 13) and other (tumours with no identifiable recurrent molecular feature, *n* = 15). Truncating mutations include nonsense mutations, insertions or deletions that alter the reading frame and splice-site mutations. *Indicates a *FH* missense mutation (G401V), likely a passenger mutation. Percentages on the left indicate frequencies of 4 distinct subsets within the uRCC cohort. Percentages on the right indicate mutation frequencies of corresponding gene(s) within the cohort. **(b)** Progression-free survival (left) and cancer-specific survival (right) associated with NF2 loss, mTORC1, FH, chromatin DNA damage and other groups are presented and colour-coded as in **a**. Statistical significance was determined by log-rank test. Statistical significance is indicated as ***P < 0.001; **P < 0.01; *P < 0.05.

accounted for 55% of our uRCC cohort (Fig. 4a). Of the remaining 28 (45%) uRCC, 8 cases carried mutations of genes involved in chromatin modulation (*SETD2, BAP1, KMT2A/C/D* and *PBRM1*); 5 in DNA damage response (*TP53, CHEK2* and *BRCA2*); and 15 without recurrent molecular features based on our analyses (Fig. 4a). The possibility of these tumours representing other RCC subtypes (for example, TFE3/TFEB translocation or SDHB deficiency) was also excluded by established diagnostic assays[46,47].

Commonly mutated in VHL-deficient ccRCC[21], chromatin modulators *PBRM1, SETD2* and *BAP1* were recurrently mutated in uRCC that lacked *VHL* mutations. Our finding that these mutations also recur in nccRCC is in line with the recently reported mutations of SWI/SNF and chromatin modifier pathways in type 1 and type 2 pRCC[16]. Given the presence of mutations of chromatin modulation or DNA damage response genes in a wide variety of cancers and their known implications in tumorigenesis[48,49], we tentatively grouped together the uRCC

cases with mutations in these pathways and lacking other apparent driver alterations.

Among the 15 cases lacking recurrent features ('other' group), T62 and T69 had non-recurrent *MET* (H1094Y) or *BRAF* (Y472C) pathogenic mutations, respectively (Supplementary Data 2)[50,51]. Together, there were seven cases in which no mutation or other significant molecular alteration was detected by our panel of analyses, but the clinicopathologic features of these cases (for example, high-grade nuclear features, necrosis and so on) excluded the possibility of them being reclassified as renal oncocytomas. In addition, three uRCC tumours with somatic *SMARCB1* mutations (T23, T38 and T41) retained the INI1 protein expression (encoded by *SMARCB1*), and were histologically distinct from renal medullary carcinoma that exhibits characteristic INI1 loss and occurs in individuals with sickle cell trait or other hemoglobinopathies[52] (Supplementary Fig. 7). These three tumours were assigned into different molecular subsets (that is, mTORC1, NF2 loss or FH) based on their other aberrations.

**Differential clinical outcomes observed in molecular subsets.** Despite the relatively small patient sample size of our high-grade uRCC cohort, differential cancer-specific outcomes were observed among the above-defined molecular subgroups (Fig. 4b). NF2 loss and FH-deficient uRCC appeared to have worse clinical outcome than mTORC1 hyperactive and thus far unspecified uRCC, whereas uRCC with mutations mainly in chromatin modulation or DNA damage response genes fared intermediately (Fig. 4b).

SETD2 or BAP1 mutation alone did not discern tumour subsets with significantly different clinical outcomes in this uRCC cohort (Supplementary Fig. 8).

## Discussion

This study presents the first in-depth molecular characterization of high-grade uRCC, a rare and heterogenous group of aggressive tumours that poses one of the most important diagnostic and therapeutic challenges among rare kidney cancers. Our integrated, step-wise, molecular approach yields molecularly distinct subsets accounting for ~76% of the uRCC cohort, and we are able to show differential clinical outcomes associated with these molecular subsets. We identified 29 recurrently mutated genes (Fig. 1) including NF2 (18%), SETD2 (18%) and BAP1 (13%) as the most frequently mutated genes. Although some of these mutations are present in certain established subtypes of RCC, the overall mutation profiles, the frequencies of mutations in specific genes and a lack of characteristic molecular features of established RCC subtypes support the notion that these uRCC tumours are largely distinct from the established RCC subtypes and harbour their unique oncogenic pathways.

This study identifies a subset of uRCC that is characterized by NF2 loss, dysregulated Hippo–YAP signalling and aggressive clinical behaviour (Figs 2 and 4). The majority (69%) of this subset demonstrates biallelic inactivation of NF2 with concurrent NF2 mutation and LOH, a molecular feature that has not been reported in RCC. In the remaining cases with only LOH of NF2, low NF2 protein levels were observed. While NF2 has been shown to be a haploinsufficient tumour suppressor in mice[24], it is possible that other mechanisms further inactivate its function in these cases. As the regulation of Hippo signalling could differ based on organ or cellular contexts[53], the YAP activation we observed predominantly in the NF2 loss subset of uRCC suggests that NF2 inactivation is an essential mechanism dysregulating Hippo signalling in RCC. The high prevalence of NF2 loss in a distinct subset of uRCC suggests it acting as an early driver event in the tumorigenesis, although this remains to be further investigated. Other molecular features found in this subset of tumours include the enrichment of SETD2 mutations, frequent 1p and 3p losses and aberrant histone methylation (absence of H3K36me3) in cases with concurrent 3p loss and SETD2 mutation. The identification of this NF2 loss subset of uRCC provides an opportunity to improve our diagnosis of this particularly aggressive subset of tumours, and test new therapeutic strategies such as those aimed at interfering with YAP activity[54], or the synthetic lethal interaction of WEE1 inhibition in H3K36me3-deficient cancer[55].

The mTORC1 hyperactive uRCC displayed much higher levels of mTORC1 signalling than the other uRCC, and was associated with a better clinical outcome. This subset harbours molecular alterations similar to those identified in a small cohort of ccRCC patients who benefit long term from mTOR inhibitor therapy[56], and suggests a readily available targeted therapy venue for patients with advanced uRCC that belong to this subset.

We detected three HLRCC cases with proven germline FH mutations in our uRCC cohort, emphasizing the wide histological spectrum observed in HLRCC-associated renal tumours[40,42]. We also found one case harbouring somatic alterations (homozygous deletion) that fully inactivate FH function. Together, these four FH-deficient tumours showed the worst clinical outcome in this cohort. Nonetheless, given the rare incidence of FH somatic alterations reported to this date in RCC[16], it remains unclear whether the clinical behaviour of a FH-deficient RCC due to somatic alterations resembles that of HLRCC. The current histological criteria and immunohistochemical markers for FH deficiency (FH and 2SC) appear to be insufficient to reliably distinguish the latter tumour from true HLRCC cases, and it remains critical to recommend genetic counselling when pathologic assessment raises a suspicion of HLRCC.

We also identified a tumour with TPM3–ALK fusion, a second RCC case with this specific fusion reported in adults. Identifying these specific driver events supports dissecting them out from the uRCC category to facilitate future characterization of these emerging RCC subtypes.

Similar to what have been described in ccRCC and pRCC[10,11,20,15–17], mutations in chromatin modulation genes are relatively frequent in uRCC, although none of which (for example, SETD2 and BAP1) was found to be significantly associated with clinical outcomes in this cohort. We did not observe specific patterns of distribution for these mutations, except for the enrichment of SETD2 mutations in the NF2 loss subset. Further validation studies are needed to clarify the roles of these mutations in the oncogenesis of various types of RCC.

MET mutations have been predominantly, but not exclusively detected in pRCC based on the recent genomic studies of RCC[10,15–17]. While the discovery of MET H1094Y mutation in one uRCC may suggest it represents a pRCC with atypical histologic features, more importantly it provides a potential therapeutic option for this patient.

Based on this molecular study of uRCC, it is tempting to speculate that NF2 loss, mTORC1 hyperactivity, FH deficiency and chromatin modulation/DNA damage defects could play key roles in the tumorigenesis and help explain the highly heterogeneous nature of uRCC. In conclusion, this study not only demonstrates shared molecular features between uRCC and other RCC subtypes, but also uncovers unique oncogenic pathways to uRCC, which could have future diagnostic, pathogenic and therapeutic implications for rare kidney cancer patients.

## Methods

**Human tumour samples.** Frozen or FFPE tissue samples were collected from primary nephrectomy specimens performed at Memorial Sloan Kettering Cancer Center (MSKCC) and processed according to MSKCC institutional review board approved tissue collection protocols with informed consent from all patients. The study was approved by our institutional review board. All cases have been reviewed and confirmed to fulfill the diagnostic criteria of renal cell carcinoma, unclassified (uRCC) by three genitourinary pathologists (Y.B.C., V.E.R. and S.K.T.) based on the current World Health Organization classification and concensus diagnostic criteria of International Society of Urological Pathology[2,3]. Ancillary studies such as TFE3/TFEB immunohistochemistry and FISH were performed to exclude tumours harbouring MiTF family translocations. Tissues were macro-dissected to ensure >70% tumour content. DNA was extracted from tumour or matched normal tissue using DNeasy Blood and Tissue kit or QIAamp DNA FFPE Tissue kit (Qiagen) for frozen or FFPE tissue, respectively, according to the manufacturer's instructions. Total RNA was purified from tumour and normal tissues using mirVana miRNA Isolation kit (Ambion) according to the manufacturer's instruction for total RNA isolation. DNA from each sample was quantified using Qubit fluorometer (ThermoFisher). RNA from each sample was analysed using Bioanalyzer assay (Agilent Technologies).

**Targeted sequencing and analysis.** The IMPACT assay is a next-generation sequencing platform that uses hybridization-based exon capture and massively parallel DNA sequencing to capture all protein-coding exons and selected introns of oncogenes, tumour suppressor genes and members of pathways deemed actionable by targeted therapies[18,19]. It is suitable for DNA extracted from either

fresh-frozen or FFPE samples. The assay used for this study included 230 key cancer-associated genes (Supplementary Data 1). In brief, barcoded sequences were prepared and captured by hybridization with custom biotinylated DNA probes for all exons and selected introns of these 230 genes using 100–500 ng of input DNA. Captured libraries were sequenced on an Illumina HiSeq ($2 \times 100$ bp paired-end reads). The raw reads were aligned to the human genome (hg19) using Burrows-Wheeler Alignment Tool (BWA-MEM), followed by duplicate read removal, base recalibration and indel realignment using GATK (v 2.6–5)[57]. We obtained an average sequence coverage depth of $348 \times$ per tumour and $280 \times$ per normal samples. Somatic variants were called using MuTect (v 1.1.4)[58] for single-nucleotide variants and Somatic Indel Detector (GATK 2.3–9) for indels, and annotated by Annovar for cDNA and amino-acid changes as well as presence in dbSNP database (v137) and COSMIC database (v68) and 1000 Genomes minor allele frequencies. Copy number was computed using tumour:normal ratios of normalized coverage data to determine amplifications and deletions except for data on chromosome X[18]. IMPACT was designed to focus on somatic mutation detection by filtering out alterations also present in matched normal samples. Matching normal was available for 61 out of 62 uRCC cases and the normal DNAs were sequenced in parallel with the corresponding tumour DNAs. For the remaining case (T02), a randomly selected normal DNA sample was used as unmatched normal control.

**SNP array analysis of the tumour genome.** Genome-wide DNA copy-number alterations and allelic imbalances were analysed by SNP array using Affymetrix OncoScan FFPE Assay (Affymetrix). We used 80 ng of genomic DNA extracted from FFPE tissue for each tumour sample. The samples were processed according to the manufacturer's guidelines. In brief, genomic DNA was annealed to MIP probes, followed by gap filling, ligation, digestion, amplification and hybridization to the microarrays using the Affymetrix GeneChip 3000 System. The data were analysed by the OncoScan Console (Affymetrix) and Nexus Express (BioDiscovery) softwares using Affymetrix TuScan algorithm. All array data were also manually reviewed for subtle alterations not automatically called by the software.

**RNA-seq and GSEA.** RNA-seq libraries were prepared using the TruSeq RNA Sample Preparation kit (Illumina) according to the manufacturer's protocol. Libraries were sequenced on the Illumina HiSeq2500 platform to obtain on average, 80 million paired-end ($2 \times 75$ bp) reads per sample. Sequence data were processed and mapped to the human reference genome (hg19) using STAR (v2.3)[59]. Gene expression levels were quantified with HTSeq-count[60] and normalized using DEseq[61]. We used GSEA[62] to statistically evaluate pathway or gene set activity that may associate with NF2 status. We tested YAP/TAZ targeted gene sets (differentially expressed genes between si-YAP/TAZ and non-targeting control) derived from the previous study[33] in the GSEA analysis (Supplementary Data 3). We first removed genes expressed at low levels in all tumours (read count $< 20$ in all samples) from the analysis. The expressed genes ($= 16,658$) were tested for differential expression between the NF2 mutated ($n = 4$) and wild-type ($n = 3$) samples with RNA-seq data available. We used the voom method for differential expression[63], which applies an empirical Bayes approach suitable for small sample sizes, to compute a moderated $t$-statistic for the null hypothesis that there is no difference in expression between the two groups. Genes were sorted by this $t$-statistic, and GSEA was used to evaluate the null hypothesis that genes in selected gene sets were not differentially expressed in mutated versus wild-type samples (using 1,000 permutations).

**Immunohistochemistry.** Immunohistochemistry was conducted in 5 µm FFPE whole tissue or tissue microarray sections using automated Ventana Discovery system or Ventana Benchmark system (Ventana Medical Systems). The primary antibodies used included NF2 (1:100, D3S3W, Cell Signaling Technology), YAP/TAZ (1:50, D24E4, Cell Signaling Technology), phospho-YAP (Ser127) (1:500, D9W2I, Cell Signaling Technology), phospho-S6 (Ser235/236; 1:100, D57.2.2E, Cell Signaling Technology), phospho-4BP1 (Thr37/46; 1:400, 236B4, Cell Signaling Technology), 2SC (Dr Norma Frizzell, Univ. of South Carolina)[42], FH (1:1,000, Clone J-13, Santa Cruz Biotechnology), INI1 (1:100, BAF47, BD Bioscience) and H3K36me3 (1:200, MABI-0333, Active Motif). For the semi-quantitative or quantitative (H-scores) analysis of staining, the pathologists were blinded to the group designation of cases on tissue microarray slides.

**Fluorescence in situ hybridization.** NF2/22q FISH analysis was performed on paraffin section (5 µm) using a three-color probe mix as described in Supplementary Table 2. Clone DNA was labelled by nick translation using fluorochrome-conjugated dUTPs from Enzo Life Sciences Inc., supplied by Abbott Molecular Inc. Hybridization, post-hybridization washing and fluorescence detection were performed according to standard procedures. Slides were scanned using a Zeiss Axioplan 2i epifluorescence microscope equipped with a megapixel charge-coupled device camera (CV-M4$^+$CL, JAI) controlled by Isis 5.2 imaging software (Metasystems Group Inc, Waltham, MA). The entire section was scanned under $\times 63$ objective to assess copy-number change and possible intratumoral heterogeneity. Representative regions were imaged through the depth of the tissue (compressed/merged stack of 12 z-section images taken at 0.5 µm intervals under

the red, green and orange filter, respectively). A minimum of two to three tumour image fields ($> 100$ cells) were selected and the total number of signals scored for each locus. Non-tumour area(s) or normal tissue including stromal cells or infiltrating lymphocytes were also analysed and served as the internal control to assess quality of hybridization. A minimum of 100 non-tumour cells were also scored.

Interphase FISH analysis on FFPE tumour tissue was perform to evaluate ALK gene rearrangements, using dual-colour break-apart probes targeting ALK gene (Abbott Molecular). The orange fluorochrome direct labelled probe hybridizes distal (3′) to ALK gene; the green fluorochrome direct labelled probe hybridizes proximal (5′) to ALK. Nuclei were counterstained with 4,6-diamidino-2-phenylindole (blue), and interphase FISH signal scoring was performed. In a normal interphase nucleus, two orange/green fusion signals are expected. Signals for ALK gene rearrangement are either 'broken apart' signal or 'single orange' signal (deleted green signal for 5′ALK). One-hundred interphase cells from the area with high tumour content were analysed.

**Plasmids.** Flag-tagged mTOR (pcDNA3-Flag-mTOR wt) was a gift from Jie Chen (Addgene plasmid # 26603). HA-GST-tagged S6K1 (pRK5-HA-GST-S6K1) was a gift from Dr David Sabatini. The mTOR single mutations were generated by introducing corresponding nucleotide changes into pcDNA3-Flag-mTOR using QuikChange II XL site-directed mutagenesis kit (Agilent). All the constructs were confirmed by DNA sequencing. The primers for site-directed mutagenesis are as follows: mTOR L2427R, 5′-CATCAGCCTCCAGTTCCGCAAGGGGT CATAGAC-3′; mTOR V2475M, 5′-AATAGATTCTGGCATTGTGGTCCCC GTTTTCTTATGGG-3′.

**Short hairpin RNA-mediated knockdown.** pLKO1-shYAP1_1 was a gift from Kunliang Guan (Addgene plasmid # 27369). pLKO1-shYAP1_2 was obtained from Sigma-Aldrich (TRCN0000107266). Lentiviral vectors carrying the indicated short hairpin RNA were co-transfected with pCMVΔR8.2 and pHCMV.VSVG into 293T cells to generate lentivirus. ACHN and LB996-RCC cells infected with lentivirus were under puromycin selection at 2 and 1 µg ml$^{-1}$, respectively.

**Cell culture and in vitro assays.** 293T cells were cultured in DMEM (Invitrogen) supplemented with 10% fetal bovine serum (FBS), non-essential amino acids, L-glutamine, sodium pyruvate and antibiotics (Invitrogen). ACHN cells were cultured in RPMI 1640 (Invitrogen) supplemented with 10% FBS, non-essential amino acids, L-glutamine, sodium pyruvate and antibiotics (Invitrogen). LB996-RCC cells were cultured in IMDM (Invitrogen) supplemented with 10% FBS, GlutaMAX, G-5 supplements and antibiotics (Invitrogen). The ACHN cells were provided by the National Cancer Institute Developmental Therapeutics Program (Bethesda, MD), and the LB996-RCC cells were provided by Dr Van den Eynde (Ludwig Cancer Research Center, Brussels), whose laboratory established this cell line[64]. To assay cell proliferation, $1 \times 10^5$cells were seeded onto a 35 mm dish and counted 4 days later. To assay cell cycle, cells were trypsinized, washed with PBS, treated with 20 µg ml$^{-1}$ RNase A, and stained with 25 µg ml$^{-1}$ propidium iodide (PI) for 1 h before being subjected to cell cycle analyses. Flow-cytometric analyses were performed using a FACSCalibur flow cytometer (Becton-Dickinson) to measure DNA contents. And, data were analysed with FlowJo software (Tree Star). To perform soft agar assay, $1 \times 10^5$ cells were seeded onto a 6 cm dish containing a top layer of 0.3% noble agar and a bottom layer of 0.6% noble agar base. Cells were fed with media every 3 days. After 3 weeks, colonies with diameter $> 200$ µm were scored. Three independent triplicate experiments were performed. For mTORC1 signalling experiments, 293T cells were seeded in 6-well plates ($1.8 \times 10^6$ cells per well) 24 h before transfection by 1.5 µg of vectors expressing wild type or mutant mTOR and 50 ng of vector expressing S6K using Lipofectamine 2000 (Invitrogen). The cells were collected 48 h post transfection.

**Protein blot analysis.** Cells were collected in ice-cold PBS buffer, pelleted and lysed in RIPA buffer (150 mM NaCl, 1% NP-40, 1% Na deoxycholate, 0.01 M Sodium phosphate (pH 7.2), 0.1% SDS, 2 mM EDTA and 50 mM NaF) with complete protease inhibitor (Roche) and phosphatase inhibitors (EMD/Millipore). Protein concentration was determined by the BCA kit (Pierce). Protein samples (20–40 µg) were resolved by 4–12% NuPAGE (Life Technologies), transferred onto polyvinylidene difluoride membrane (Immobilon-P, Millipore) and detected by the enhanced chemiluminescence method (Western Lightning, PerkinElmer) and LAS-3000 Imaging system (Fujifilm). The blot images were analysed by Image-Gauge software (Fujifilm).

Antibodies used for immunoblot analysis are as follows: anti-NF2 (ab88957, Abcam), anti-YAP1 (no. 12395, Cell Signaling Technology), anti-pSer-127 YAP1 (no. 13008, Cell Signaling Technology), actin (A1978, Sigma), anti-pThr-389 S6K (no. 9205, Cell Signaling Technology), anti-HA (12CA5), anti-pSer-65 4EBP1 (no. 9451, Cell Signaling Technology), anti-4EBP1 (no. 9452, Cell Signaling Technology), anti-pSer 235/236 S6 (no. 4858, Cell Signaling Technology), anti-S6 (no. 2217, Cell Signaling Technology), anti-Flag (F1804, Sigma), anti-RAPTOR (no. 2280, Cell Signaling Technology). The dilution for all of the primary antibodies for immunoblot analysis was 1/1,000. The dilution for all of the secondary antibodies for immunoblot analysis was 1/5,000. The full blots from these analyses are shown in Supplementary Fig. 9.

**Statistical analysis.** Significant co-occurrence or mutual exclusivity was determined using Fisher's exact test. Statistical significance of quantified or semi-quantified immunohistochemical staining between tumour groups was determined by Mann–Whitney $U$-test. Statistical significance of cell line experiments was determined by Student's $t$-test. Significance of survival curves was analysed using the log-rank test.

**Data availability.** The RNA-seq and Oncoscan SNP array data have been deposited in the database of Gene Expression Omnibus under accession GSE85971. The IMPACT targeted DNA sequencing and clinical data of the cohort are available for public access at cBioPortal (http://www.cbioportal.org/study?-id=urcc_mskcc_2016#summary)[65]. Any other data are contained within the article and its Supplementary Information files or available from the authors on request.

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

## Acknowledgements

We are grateful to all the patients who agreed to participate in this study and the physicians who have provided their medical care. We thank A. Bialik, Y. Xiao and M. Asher for technical assistance. We thank A. Viale and the personnel of the Genomics and Bioinformatics cores for assistance with sequencing and data analysis. This work was supported by the Cycle for Survival Fund and the Society of Memorial Sloan Kettering research grant (Y.B.C.), the MSKCC Pathology, Genomics, and Molecular Cytogenetics cores (core grant P30 CA008748), the US National Institutes of Health (NIH) R01 CA152975 (F.G.G.), and The Tuttle Family Rare Kidney Cancer Fund (J.J.H.).

## Author contributions

Y.B.C., V.E.R., M.F.B. and J.J.H. designed and oversaw the study. Y.B.C., S.K.T. and V.E.R. performed histological assessment. A.J.S., A.R.B., H.H.W., P.I.W., W.L., N.S. and M.F.B. analysed DNA and RNA-seq data. L.W. analysed SNP array data. A.A.J. performed H3K36me3 immunohistochemical study. Y.B.C. analysed the immunohistochemistry data. G.J.N. and L.W. performed FISH studies. J.X., Y.D. and V.O. performed the cell line experiments. F.G.G., E.H.C., Y.B.C. and J.J.H. designed the cell-based assays. W.L. and F.G.G. provided NF2-related gene expression signatures. A.A.H., M.H.V., R.J.M. and P.R. recruited patients and provided clinical database. Y.B.C. and J.J. H. drafted the manuscript. All authors contributed to critical review of the paper.

## Additional information

**Competing financial interests:** The authors declare no competing financial interests.

