## [Peer Review File · Nature Communications]

Reviewers' comments:

Reviewer #2 (Remarks to the Author):

The paper provides a genomic description of 62 high-grade primary unclassified RCCs, a rare type of renal cell carcinoma. Amongst the key findings are a subset (26%) characterized by NF2 loss, a subset with hyperactive mTORC1 and an FH deficient subset.

A genomic description of unclassified nccRCC is much needed and due to its rarity a description of 62 samples is significant.

Overall the data and methods used are solid. One main concern on the results is that the authors appear not to reconsider the original diagnosis provided by pathology.

When mutation results point to events that one could use to reconsider the pathology, the original pathology annotation as unclassified remains. The reasoning on why to stick with the original pathology is not well discussed in the paper.

For example why were the samples with FH germline mutations not reclassified as HLRCC? Should these samples be reclassified and mentioned in the main text? Figure S4 suggests in the comment column HLRCC. If reclassification seems warranted, the FH-deficiency subgroup in uRCC would be actually be mislabeled HLRCC and not a uRCC subgroup.

The two samples with Met mutations could possibly be pRCC given that Met has been reportedly mutated in 15% of pRCC. Has this been considered?

Was the RNA-Seq data used to look for fusion events including MITF, TFEB and TFE3? These fusions have been found in nccRCC samples originally classified as uRCC but are likely tRCC.

Can a renal oncocytoma reclassification of the samples where no mutations were found be excluded?

Was RNA-Seq data obtained for all samples, and if so should an expression based sample clustering be considered? This could ideally be done including one of the recently published nccRCC expression datasets that include pRCC, chromophobe and tRCC samples to look for samples that cluster with those histologies.

Figure 2h should indicate which colony picture is YAP1 or control GFP.

Reviewer #3 (Remarks to the Author):

In this manuscript the authors analyzed unclassifiable Renal Cell Carcinoma, a RCC category for which the oncogenic drivers have not been defined. The molecular understanding of other, common RCC subtypes has been well established, and this manuscript adds important molecular information to kidney cancer. Management and diagnosis of unclassifiable RCC is difficult and improved understanding of molecular drivers may have diagnostic and therapeutic consequences.

The research is well designed and the conclusion that uRCC can be subdivided into distinct subsets

is substantiated by their analysis. In only few cases obvious driver gene signatures are absent. Comments:

The description of the patient population is somewhat confusing. 36 patients were diagnosed with locally advanced RCC, but the manuscript describes these as "developed metastatic disease", suggesting that patients were initially diagnosed as M0. In fact 26 patients were M1, i.e., 10 patients developed metastases after nephrectomy.

The observation that 22/26 patients died of RCC is more a reflection of poor therapeutic possibilities than aggressive clinical behavior.

Reviewer #4 (Remarks to the Author):

Author: Chen et al.

Title: Molecular analysis of aggressive renal cell carcinoma with unclassified histology reveals distinct subsets associated with varying clinical outcomes

Comments:

A. Summary of the key results:

In this manuscript, Chen et al tried to identify the oncogenic drivers for the unclassified histology renal carcinoma (uRCC) by molecular analysis of 62 high-grade primary uRCC. Based on this analysis, they classified 76% of uRCC cohort into 4 subtypes: NF2 loss (26%), hyperactive mTORC1 (21%), FH deficiency (6%), and chromatin/DNA damage regulator mutations (21%) and ALK-translocation (2%). They claimed that it could have diagnostic and therapeutic implications.

B. Originality and interest:

Although this is the first in-depth molecular characterization of uRCC, Hippo and mTOR have already reported to be involved in RCC. It was also recently reported that alterations of the Hippo pathway and mTOR are detected in other subtypes of RCC (Chen et al., Cell Rep, 2016). Therefore, alterations of Hippo or mTOR can not specifically used for classification of uRCC subtypes.

C. D. Data & methodology & appropriate use of statistic and treatment of uncertainty

1) Fig. 2e: the data for quantification of (p)YAP/TAZ subcellular localization in tumors is not convincing. It is unclear why it is statistically significant if the standard deviations (S.D) of YAP/TAZ (nuclear) staining between NF2-loss and other uRCC are overlapping. In addition, YAP/TAZ are usually localized in both nucleus and cytoplasm. However, there is no sample showing this type of localizations. Moreover, due to heterogeneity of tumor tissues, only a proportion of tissues may have the stated staining phenotype (nuclear vs cytoplasm). Therefore, H-score instead of IHC score number alone should be used for quantification.

2) Fig. 2g:

a) Due to off-target effect of shRNA, at least two shYAPs should be used in functional analysis.

b) ACHN cells were reported to bear a deletion of Sav, a downstream target of NF2 on the hippo pathway (Tapon et al., 2002). Therefore, the tumorigenic phenotype may not be caused by loss of NF2, as stated in this manuscript. In addition, it has already been reported that inactivation of YAP in this cell line inhibit its tumorigenic phenotypes such as loss of cell-cell contact inhibition (Zhao et al., 2007).

E. Conclusion:

The sample number is too low to get any conclusion for diagnosis purpose. For example, for FH deficiency subtype, only 3 uRCC patients were detected with FH mutations.

F. Suggested improvement

See above comments

G. Reference: OK

H. Clarity and context: OK

We thank reviewers' positive and helpful comments. As suggested, we have performed the recommended experiments/analyses, and revised the manuscript to clarify and further strengthen our findings. We have revised **Figure 2 (e and g)** and added a new **Supplementary Figure 2** to display data from the new analysis and *in vitro* cell line experiments. We believe that we have adequately addressed the comments from the reviewers. Below is our point-by-point response to the reviewers' comments.

Reviewers' comments:

Reviewer #2:

A genomic description of unclassified nccRCC is much needed and due to its rarity a description of 62 samples is significant.

Response: We thank the reviewer for pointing out the critical need for a genomic analysis of unclassified nccRCC. As commented by the reviewer, the 62 cases that we assembled for this study represent a significant effort to molecularly characterize unclassified nccRCC, a rare and heterogeneous group of tumors.

Overall the data and methods used are solid. One main concern on the results is that the authors appear not to reconsider the original diagnosis provided by pathology.

When mutation results point to events that one could use to reconsider the pathology, the original pathology annotation as unclassified remains. The reasoning on why to stick with the original pathology is not well discussed in the paper. For example why were the samples with FH germline mutations not reclassified as HLRCC? Should these samples be reclassified and mentioned in the main text? Figure S4 suggests in the comment column HLRCC. If reclassification seems warranted, the FH-deficiency subgroup in uRCC would be actually be mislabeled HLRCC and not a uRCC subgroup.

Response: We thank the reviewer's expert opinion on our data and methods. The goal of the molecular analysis described in this study is to help dissect distinct subsets within the unclassified RCC group, so we entirely agree that certain mutation results, such as *FH* germline mutations, can be used to re-classify the tumors. To clarify this point, we have modified the corresponding text in "Results" and "Discussion" sections (changes of text are in **bold** fonts):

“Genetic testing revealed *FH* germline mutations in 3 of these 4 patients, **confirming that they indeed represent HLRCC cases.** The remaining *FH* negative/2SC positive tumor (T41) harbored somatic homozygous deletion of the *FH* gene, **revealing a somatic mechanism that can lead to *FH* functional loss.**”

“The *TPM3-ALK* fusion has been reported in human cancers⁴⁴, including in one uRCC case⁴⁵. **Reported rarely in children and adults, ALK translocation-associated RCC is currently considered as an emerging entity, awaiting further characterization³.**”

“We detected 3 HLRCC cases with proven germline *FH* mutations in our uRCC cohort, emphasizing the wide histological spectrum observed in HLRCC-associated renal tumors^{40,42}. **We also found one case harboring somatic alterations (homozygous deletion) that fully inactivate FH function.** Together, these 4 FH-deficient tumors showed the worst clinical outcome in this cohort. **Nonetheless, given the rare incidence of *FH* somatic alterations reported to this date in RCC¹⁶, it remains unclear whether the clinical behavior of a FH-deficient RCC due to somatic alterations resembles that of HLRCC.** The current histological criteria and immunohistochemical markers for FH-deficiency (FH and 2SC) appear to be insufficient to reliably distinguish the latter tumor from true HLRCC cases, and it remains critical to recommend genetic counseling when pathologic assessment raises a suspicion of HLRCC.

We also identified a tumor with *TPM3-ALK* fusion, a second RCC case with this specific fusion reported in adults. **Identifying these specific driver events supports dissecting them out from the uRCC category to facilitate future characterization of these emerging RCC subtypes.**”

The two samples with Met mutations could possibly be pRCC given that Met has been reportedly mutated in 15% of pRCC. Has this been considered?

Response: There is one case (T62) with MET (H1094Y) mutation. During the pathologic re-review, we did consider the possibility of papillary RCC (pRCC) since the tumor showed papillary architecture in areas. However, it also showed features not typical for pRCC, such as multinodular infiltrating growth, and marked clear cell morphology in areas. We could not be certain that it belongs to pRCC group and retained the original pathologic diagnosis. Although predominantly detected in pRCC based on the recent genomic studies of RCC, rare MET mutations have also been reported in TCGA clear cell RCC (ccRCC) study. We have added the below sentence in the “Discussion” section to address this point:

“***MET* mutations have been predominantly but not exclusively detected in pRCC based on the recent genomic studies of RCC^{10, 15-17}. While the discovery of *MET* H1094Y mutation in one uRCC may suggest it represents a pRCC with atypical histologic features, more importantly it provides a potential therapeutic option for this patient.**”

Was the RNA-Seq data used to look for fusion events including MTF, TFEB and TFE3? These fusions have been found in nccRCC samples originally classified as uRCC but are likely tRCC.

Response: Since MiTF translocation related RCC is a recognized distinct family of RCC, during our initial assembly of the cohort, we screened all cases by TFE3 and TFEB immunohistochemistry, and if cases showed any positivity or equivocal staining, they were also studied by TFE3 and/or TFEB FISH. All fusion positive cases were excluded from the

study. Therefore, the inclusion of MiTF family translocation associated RCC in this cohort is very unlikely. The RNA-seq analysis was only performed in a subset of 7 cases, and we did not detect any positive fusion events involving MiTF, TFEB or TFE3.

We have added a sentence in the “Human tumor samples” section of the “Methods” to clarify this point. **“Ancillary studies such as TFE3/TFEB immunohistochemistry and fluorescence in-situ hybridization were performed to exclude tumors harboring MiTF translocations.”**

Can a renal oncocytoma reclassification of the samples where no mutations were found be excluded?

Response: Although there was no mutation detected in 9 cases by our panel of analysis, the clinicopathologic features of these cases (e.g. high grade nuclear features, necrosis etc.) excluded the possibility that they would be reconsidered as renal oncocytoma. We have added this sentence to the “Results” section: **“Together there were 9 cases in which no mutation was detected by our panel of analyses, but the clinicopathologic features of these cases (e.g. high grade nuclear features, necrosis etc.) excluded the possibility of them being reclassified as renal oncocytomas.”**

Was RNA-Seq data obtained for all samples, and if so should an expression based sample clustering be considered? This could ideally be done including one of the recently published nccRCC expression datasets that include pRCC, chromophobe and tRCC samples to look for samples that cluster with those histologies.

Response: We were not able to obtain RNA-Seq data for all samples, which was largely due to the limited availability of frozen tissue in these rare tumors. While an expression based clustering analysis is very important, we could not reliably perform this analysis in this cohort.

Figure 2h should indicate which colony picture is YAP1 or control GFP.

Response: We have revised Figure 2h to indicate this point.

Reviewer #3:

The research is well designed and the conclusion that uRCC can be subdivided into distinct subsets is substantiated by their analysis. In only few cases obvious driver gene signatures are absent.

Response: We greatly appreciate this reviewer’s positive comments on our study.

The description of the patient population is somewhat confusing. 36 patients were diagnosed with locally advanced RCC, but the manuscripts describes these as "developed metastatic disease ", suggesting that patients were initially diagnosed as M0. In fact 26 patients were M1, i.e., 10 patients developed metastases after nephrectomy.

Response: In Table S1, the number of patients with pT3/4 disease and the number of patients with M0 are coincidentally the same (36), which might have caused confusion. To clarify, we have added time of assessment such as “**at nephrectomy**” or “**at last follow-up**” to each parameters.

The observation that 22/26 patients died of RCC is more a reflection of poor therapeutic possibilities than aggressive clinical behavior.

Response: We agree with the reviewer that poor therapeutic possibilities available for patients who died of unclassified RCC contribute significantly to the aggressive clinical behavior observed for these patients. We have modified the following sentence in the “Results” section:

“..., underscoring the aggressive clinical behavior and **poor response to systemic therapies** observed in this uRCC cohort.”

Reviewer #4:

B. Originality and interest:

Although this is the first in-depth molecular characterization of uRCC, Hippo and mTOR have already reported to be involved in RCC. It was also recently reported that alterations of the Hippo pathway and mTOR are detected in other subtypes of RCC (Chen et al., Cell Rep, 2016). Therefore, alterations of Hippo or mTOR can not specifically used for classification of uRCC subtypes.

Response: We greatly appreciate this reviewer’s positive comments on our study, and we also thank reviewer for the comments on Hippo and mTOR pathways in RCC. Alterations in Hippo pathway have been described in rare ccRCC and pRCC cases, including in the pRCC P.CIMP-e group as summarized in the pan-TCGA RCC study by Chen F. *et al* (Cell Rep, 2016). However, the exact role of this pathway as it relates to other key driver events in ccRCC and pRCC are unclear. Here we describe a subset of uRCC with aberrant Hippo pathway while lacking molecular features of established subtypes of RCC, suggesting it may be exploited as a targetable pathway in this subset of tumors. Similarly, alterations of mTOR pathway in a subset of uRCC suggest these tumors may be responsive to mTOR inhibitor treatment. So while these alterations are not unique, they revealed distinct molecular pathways that are likely at play in different subsets of uRCC. Hence, we think these molecular alterations may provide a valuable approach to stratify this heterogenous group of tumors.

C. D. Data & methodology & appropriate use of statistic and treatment of uncertainty

1) Fig. 2e: the data for quantification of (p)YAP/TAZ subcellular localization in tumors is not convincing. It is unclear why it is statistically significant if the standard deviations (S.D) of YAP/TAZ (nuclear) staining between NF2-loss and other uRCC are overlapping. In addition, YAP/TAZ are usually localized in both nucleus and cytoplasm. However, there is no sample showing this type of localizations. Moreover, due to heterogeneity of tumor tissues, only a proportion of tissues may have the stated staining phenotype (nuclear vs cytoplasm). Therefore, H-score instead of IHC score number alone should be used for quantification.

Response: We thank the reviewer's expert opinion on this issue. The initial semi-quantitative analysis we performed for YAP/TAZ and p-YAP IHC had limited data range (0-3), which caused difficulty in interpreting the difference between NF2-loss and other uRCC groups (Mann-Whitney U test, error bar, ± 1 SD). Additionally, as commented by the reviewer, YAP/TAZ staining is present in both nucleus and cytoplasm, but we did not depict the cytoplasmic signal in the original bar graph because we wished to focus on stains reflecting YAP activation (high level of YAP/TAZ in nucleus and low levels of p-YAP). Based on these suggestions from the reviewer, we have now re-quantified the IHC staining using **H-scores** and included assessment of YAP/TAZ cytoplasmic staining in the bar graph (Mean with 95% CI). Using Mann-Whitney U test, there is significant difference detected between NF2-loss and other uRCC tumors. **Figure 2e has been revised to reflect this new analysis.**

2) Fig. 2g:

a) Due to off-target effect of shRNA, at least two shYAPs should be used in functional analysis.

Response: Results from two independent shYAP1s have been added (**Figure 2g** and **Supplementary Figure S2**).

b) ACHN cells were reported to bear a deletion of Sav, a downstream target of NF2 on the hippo pathway (Tapon et al., 2002). Therefore, the tumorigenic phenotype may not be caused by loss of NF2, as stated in this manuscript. In addition, it has already been reported that inactivation of YAP in this cell line inhibit its tumorigenic phenotypes such as loss of cell-cell contact inhibition (Zhao et al., 2007).

Response: We thank the reviewer's comments regarding ACHN cell line. We have now performed knock-down experiments using two independent shYAP1s in another NF2-mutant RCC cell line, LB996-RCC, and observed a similar effect in the cell proliferation assay. The new data is included in the revised **Figure 2g** and we also have added a new **Supplementary Figure S2**.

E. Conclusion:

The sample number is too low to get any conclusion for diagnosis purpose. For example, for FH deficiency subtype, only 3 uRCC patients were detected with FH mutations.

Response: As stated by the reviewer, this study is the first in-depth molecular characterization of uRCC. Based on the analysis of 62 high-grade primary uRCC, we classified 76% of the uRCC cohort into 4 subtypes: NF2-loss (26%), hyperactive mTORC1 (21%), FH deficiency (6%), and chromatin/DNA damage regulator mutations (21%) and ALK-translocation (2%). We acknowledge that the sample number within each proposed molecular subsets is still low for diagnostic purposes, but we think it is an important first step to start stratifying these tumors. Meanwhile, although very rare, FH-deficiency (or HLRCC when mutations in germline) and ALK fusion RCCs are now being recognized as distinct or emerging subtypes of RCC, and relatively sensitive and specific ancillary tools are available to help establishing these diagnoses.

REVIEWERS' COMMENTS:

Reviewer #2 (Remarks to the Author):

Thank you for addressing my comments.

Reviewer #3 (Remarks to the Author):

I have no remaining comments after the revision and feel that the manuscript is now acceptable for publication

Reviewer #4 (Remarks to the Author):

The authors have addressed all of the concerns raised by this reviewer. It can be now published.